# Development of a Complex Intervention for the Maintenance of Postpartum Smoking Abstinence: Process for Defining Evidence-Based Intervention

**DOI:** 10.3390/ijerph16111968

**Published:** 2019-06-03

**Authors:** Caitlin Notley, Tracey J. Brown, Linda Bauld, Wendy Hardeman, Richard Holland, Felix Naughton, Sophie Orton, Michael Ussher

**Affiliations:** 1Norwich Medical School, University of East Anglia, Norwich NR4 7TJ, UK; tracey.j.brown@uea.ac.uk; 2Usher Institute, College of Medicine and Veterinary Medicine, University of Edinburgh, Edinburgh EH8 9YL, UK; Linda.Bauld@ed.ac.uk; 3School of Health Sciences, University of East Anglia, Norwich NR4 7TJ, UK; w.hardeman@uea.ac.uk (W.H.); f.naughton@uea.ac.uk (F.N.); 4Leicester Medical School, University of Leicester, Leicester LE1 7RH, UK; rch23@leicester.ac.uk; 5Division of Primary Care, University of Nottingham, Nottingham NG7 2RD, UK; sophie.orton@nottingham.ac.uk; 6Population Health Research Institute, St George’s, University of London, London WC1E 7HU, UK; mussher@sgul.ac.uk; 7Institute for Social Marketing and Health, University of Stirling, Stirling FK9 4LA, UK

**Keywords:** tobacco smoking relapse prevention, postpartum women, intervention development, mixed methods

## Abstract

Relapse to tobacco smoking for pregnant women who quit is a major public health problem. Evidence-based approaches to intervention are urgently required. This study aimed to develop an intervention to be integrated into existing healthcare. A mixed methods approach included a theory-driven systematic review identifying promising behaviour change techniques for targeting smoking relapse prevention, and qualitative focus groups and interviews with women (ex-smokers who had remained quit and those who had relapsed), their partners and healthcare professionals (*N* = 74). A final stage recruited ten women to refine and initially test a prototype intervention. Our qualitative analysis suggests a lack, but need for, relapse prevention support. This should be initiated by a trusted ‘credible source’. For many women this would be a midwife or a health visitor. Support needs to be tailored to individual needs, including positive praise/reward, novel digital and electronic support and partner or social support. Advice and support to use e cigarettes or nicotine replacement therapy for relapse prevention was important for some women, but others remained cautious. The resulting prototype complex intervention includes face-to-face support reiterated throughout the postpartum period, tailored digital and self-help support and novel elements such as gifts and nicotine replacement therapy (NRT).

## 1. Introduction

Relapse to tobacco smoking following a quit attempt is a serious problem, but is often overlooked and support is under resourced. Service delivery in the UK is incentivised to achieve ‘4-week quits’, but patients are not routinely followed up beyond 12 weeks, thus those who relapse to smoking are ‘lost’ to the healthcare system and do not receive follow up support. For women who quit smoking during pregnancy, relapse is a particular concern. Relapse rates in this population are much higher than in the general population [1], perhaps due to a motivation to quit for the pregnancy alone [2], societal expectations [3], the stress and strain of new motherhood [3] or a lack of ongoing support to stay quit from smoking. Living with a partner who is a smoker is the leading predictor of postpartum relapse [3], and in our related work we theorise the important role of social identity as impacting on relapse status [4,5].

Despite the high incidence of relapse, evidence from a recent Cochrane review [6] did not support the use of specific behavioural treatments for relapse prevention, but suggested that promising treatment might include extending the use of stop smoking medication. In the UK there is currently no National Institute for Clinical Excellence (NICE) guidelines to underpin service commissioning or front-line support for relapse prevention for women who quit smoking during pregnancy. As a result, as part of ‘usual care’, most women receive little or no support to stay quit from smoking. This is a substantial gap in healthcare provision, as relapse to tobacco smoking has major personal and societal healthcare implications. People who continue to smoke are likely to die at least 10 years earlier than nonsmokers [7], but stopping and staying stopped from smoking before middle age avoids more than 90% of the health risk attributable to tobacco [8]. Women who relapse to smoking have much higher chances of developing cancer, lung or heart disease [9]. There is a generational effect, as babies exposed to second-hand smoke are more likely to suffer from respiratory and other infections and are more likely to be admitted to hospital in the first 12 months of life [10]. Children exposed to smoking behaviour are much more likely to take up smoking themselves in later life [10]. Supporting sustained smoking abstinence beyond pregnancy therefore has a major preventative role to play in the long term health of the mother and her family.

As there are no recommended approaches for the prevention of postpartum smoking relapse [6], this study was designed to develop an intervention to prevent postpartum return to smoking. Previous trials have evaluated brief intervention, motivational approaches, and some have explored extended pharmacological cessation treatments [6]. Few studies have tested more complex interventions with tailored elements to suit individual women. This is clearly an important omission, since relapse to smoking is a complex and highly individualised problem, requiring a complex solution to address the support needs of women situated within their unique social and cultural contexts. In this study we therefore sought to develop an intervention, designed to be delivered alongside or as part of usual UK NHS ante and postnatal care, for future testing in a clinical trial. We sought to work directly with pregnant and postpartum women—both smokers and ex-smokers—their partners and the healthcare professionals engaged in their care. Our approach is theory-driven [11], using the ‘Capability-Opportunity-Motivation-Behaviour’ (COM-B) framework [12] to map intervention components onto determinants of postpartum relapse, targeted by behaviour change techniques (BCTs) as ‘the smallest identifiable components of interventions’ [12]. Our approach recognises the need to move beyond motivational approaches that narrowly view pregnancy as a ‘teachable moment’ for smoking cessation [13]. We also emphasise social and cultural influences on behaviour, using social identity and the process of identity change [5] as mid-range theory to bring together macro- and micro- level influences on behaviour. This paper describes our theoretically driven, person-based approach to intervention development, working with the population we are seeking to target change are required. Adopting a theory-driven approachand mapping out an approach that may be useful in developing public health interventions where complex solutions to supporting maintenance of positive behaviour enables a clear trajectory towards evaluation of theory- and evidence-based components.

## 2. Methods

The current work is a multi-component, mixed methods study, taking a person-based approach [14] to intervention development. The study was conducted over three iterative phases, each phase informing the next. Methods are reported here in line with the criteria for reporting qualitative research (COREQ) guidance [15].

Phase 1: A systematic review of behaviour change techniques included in published postpartum relapse prevention interventions (reported elsewhere [16]). Findings from phase 1 were used to define behaviour change techniques [12] underpinning potential intervention components which were used as prompts during phase 2.

Phase 2: A qualitative study using focus groups, interviews and an online questionnaire version of the interview. We sought a purposive sample of pregnant and postpartum women, seeking to gather multiple viewpoints of those who had stopped smoking for or during pregnancy [17]. Recruitment adverts were used in stop smoking services (and directly via specialist pregnancy stop smoking advisors), in health visitor clinics attended by postpartum women, at secondary care pregnancy scanning clinics and at children’s centres in one county of the UK. Face-to-face recruitment by the research team, including health professional and clinical research facilitator support in some sites, was used. We purposefully selected both those who had remained abstinent and those who had relapsed back to smoking. We also sought the views of women’s partners and ran focus groups with health professionals involved in the care of women (including health visitors, midwives and stop smoking advisors) for triangulation of data sources [18]. We aimed to recruit approximately 64 participants, seeking maximum variation in sampling characteristics, and continuing to recruit until we felt we had achieved sufficient coverage of the range of potential views such that results might be transferable to other populations [19]. Semistructured topic guides were used, and were developed in conjunction with PPI advisory group members. Topics covered broadly mapped to the ‘Template for intervention description and replication’ (TIDieR) framework for reporting interventions [20], seeking feedback from participants on the need for relapse prevention intervention (why); who should deliver the intervention; when an intervention should be initiated, what form intervention and support should take and how long support should continue for; and we also sought feedback on some specific potential intervention components, such as e cigarettes, incentives and partner support. These specific components were chosen as presenting potentially promising approaches to relapse prevention based on existing evidence for smoking cessation in the general population [21,22], and to map to specific BCTs identified as important for postpartum relapse prevention in our systematic review [16]. CN and TB—both trained qualitative researchers—undertook focus groups together or alternatively collected data via qualitative interviews. All participants were paid travel expenses and were given a £20 shopping voucher as reimbursement for their time. We also undertook interviews and focus groups with healthcare professionals involved in the care of women postpartum. We sought a purposive sample to include midwives, health visitors, children’s centre workers and stop smoking advisors.

Following completion of phase 2, we undertook a period of developing the prototype intervention. This included working with an artist and a designer to develop a brand name, visualising aspects of the intervention (logo, app logo and leaflet). We worked with a web designer/developer developing web and app content based on findings from the review and phase 2 qualitative findings. We developed a text message support system, adapting texts used in an existing pregnancy smoking cessation support programme [21] for postpartum relapse prevention support. We also developed a mock up version of a ‘relapse kit’ information box, based on the NHS ‘quit kit’ [22].

Phase 3: Person-based intervention development [14], working with individual ex-smoking women to test and retest the prototype intervention. We purposefully selected ten postpartum women, aiming again for maximum variation in sampling characteristics (age, parity and employment status), to take part in a final phase of ‘person-based’ intervention development [14]. Women responded to adverts in recruitment sites as per phase 2 or had expressed interest during phase 2 in also being involved in phase 3. This phase involved running the refined intervention developed following phase 2 as a pilot in ‘real-time’. We met with each woman four times, approximately fortnightly, collecting detailed qualitative feedback on each prototype intervention component, including, (1) the initial introduction to the intervention, (2) receiving text message support, (3) using the intervention website/app and (4) responses to the idea of a relapse prevention kit, the ‘BabyBreathe^TM^ box’ (see Figure 1). Women were interviewed at the University of East Anglia or in their own homes. Travel expenses were reimbursed and women were given a £20 shopping voucher at each data collection meeting.

Ethical approval for phases 2 and 3 of the study was granted by West Midlands-Edgbaston NHS Research Ethics Committee (ref: 17/WM/0285).

Thematic analysis [23] of all qualitative data was undertaken led by TB. As a first step, all phase 2 data were coded descriptively to inform intervention development, aided by the use of NVivo software, version 12, QSR International, Melbourne, Australia. Three transcripts were independently coded by CN as a validity check, and the coding framework was discussed and agreed before being applied systematically across the dataset. Intervention components were then visualised and developed by the research team for use in phase 3 (e.g., a prototype website was developed and prototype text messages for the text support system were written). A second stage of inductive coding developed themes arising from the data that were less descriptive, allowing ideas and perspectives of participants to be incorporated into intervention design. At this stage, data generated through interviews, focus groups and the online version of the interview were combined. Phase 3 data was also thematically analysed to inform iterative development of intervention components. A final stage of analysis coded all prototype intervention components to map behaviour change techniques underpinning each component and inform an intervention logic model [12].

## 3. Results 

For phase 2, 74 participants were recruited and took part in focus groups, individual interviews or an online version of the interview where preferred. Participant categorisations are shown in Table 1.

Data were collected according to topic guides, ascertaining the need for support, who should deliver support, what form support should take, how long it should continue for and gathering feedback on specific potential intervention components.

### 3.1. The Need for Relapse Prevention Support

Women, partners and health professionals all acknowledged that there was little support for postpartum relapse prevention, but this was very much needed and wanted:
“I think it’s after pregnancy you’re most likely to go back to smoking because you’re detached from the baby, so you feel like you can distance yourself. And I think that’s when the work needs to be done to stay smoke-free because I know I certainly could have gone back to smoking quite easily…because as a smoker, I know what that release feels like, and you’re like ‘I just want to smoke, I just feel like that again,’ and everything can be all right for half an hour”.(034, postpartum ex-smoker)

It was almost unanimously agreed that support should be positive, offering praise and recognition for the achievement of having stopped smoking. It was felt that there was a lot of support available for pregnant women to quit smoking, and the dangers of smoking in pregnancy were well known, but there was very little support for maintaining the positive behaviour change of smoking abstinence:
“I think you could frame the relapse prevention stuff in a really positive way. When you’ve already done this really difficult thing, giving up smoking, to give babies the best chance in pregnancy, and for me starting off on a positive note like that makes me more likely to, kind of, engage with information rather than, stuff about quitting starts from quite a negative standpoint a lot of the time. How bad smoking is rather than how good it is when you quit and how to help you stay quit”.(FG 006–008, postpartum ex-smoker)

Health care professionals also felt that women currently lacked relapse prevention support and that more support should be available, delivered consistently by professionals involved in the care of women throughout pregnancy and postpartum:
“Because they need that continuous drip feeding, like we’ve already said, from the midwife they see post-birth, to the GP they see at six weeks, to the health visitor that they see, or the nursery nurse that goes in and does baby massage, they need to be able to access that support”.(HCP FG 011)

### 3.2. ‘Who’ Should Introduce Support and ‘When’?

Women suggested that having the intervention introduced by a trusted health professional, as a ‘credible source’ would be most beneficial:
“hearing it from an actual professional, you know like a health professional then you know, you’re going to believe them over, I don’t know, someone else, aren’t you?”(053, pregnant ex-smoker)

Although the intervention is aimed at supporting postpartum relapse prevention, it was almost unanimously agreed that support should begin during pregnancy to acknowledge cessation and to prepare for the immediate postpartum period where women may be vulnerable to relapse:
“I think the timing is probably quite key. If it was probably at this point, I’d be happy, I’m nearly seven months, I’d probably be happy to sit for ten minutes and discuss it with somebody”.(041, pregnant relapser)

We sought feedback on a possible leaflet to introduce relapse prevention support. This was seen as a useful prompt to underpin a face-to-face discussion with a health professional (probably a midwife) during routine antenatal care appointments:
“I think at your last midwife appointment, which I think is about 32 weeks you go. It would be like your last midwife and then you’re kind of handed over to delivery—after that, that would be—obviously keep the conversation going during pregnancy but that would be the time to say look, here’s some information about staying smoking free after you’ve had the baby”.(034, postpartum ex-smoker)

### 3.3. ‘What Form’ Should Relapse Prevention Support Take?

#### 3.3.1. Online Tailored Self-Help Support

We sought feedback on the potential ideas of signposting women to a website or to downloading an app version. Electronic support that women could access in their own time as and when they needed it was the preferred way to deliver information into the postpartum period:
“I think probably a website would be better if I was given just like maybe like a small business card size thing with like a website on that I can visit then I think that would be… yeah, then I could sort of just look in my own time really. I expect a lot of people these days also I think breastfeed with their phones in their hands, looking at the news or Facebook and things”.(021, postpartum relapser)

It was important for women that all facts given were ‘evidence-based’, using the latest statistics, and with references for further information that some women actively wanted to engage with (although not all). Information on the positive benefits of staying smoke-free particularly for the baby was desired:
“that would make them then think, because obviously they talk about more stillbirth and infant deaths more now, but then that’s like trying to find what is causing it and smoking is, well, is one of the reasons”.(037, pregnant ex-smoker)

Women also wanted information specifically on second and third-hand smoke exposure risks, on support to help partners to quit smoking, objective information about infant feeding and staying smoke-free, and advice on other support available, such as nicotine replacement therapy (NRT) and e-cigarettes:
“The nicotine replacement therapy—again knowing like what’s out there I think, because until again I had that advice and I tried different things then I would have probably just tried to do it cold turkey and that’s, it works for some people but not for everybody. So again, getting the information out there, I think is really key and what works with breastfeeding and what is more child friendly or you know takes the risks away”.(045, postpartum ex-smoker)

Possible intervention content discussed included advice and information on the health benefits for women as well as babies of staying smoke-free. Women especially liked the idea of interactive elements, such as a health timeline, that could be tailored to their delivery date:
“I think if it was something like an app where you go in and erm it’s kind of one of these ‘’Give up Smoking’—it tells you how many days you’ve been given up for and it tells you what the benefits are of what you’ve gained from having given up for that period and things like that, and it’s got little visual graphs of, you know, the positives and things like that—then it’s something that I’d probably pop into and have a look”.(041, pregnant relapser)

Women wanted links and resources that were approved and evidence-based for further information. They did not want to be ‘bombarded’ with too much information, but to have the resources available to access should they choose to do so:
“the app where they can, where somebody could go and specifically look to see what the harmful risks, and what the dos and don’ts are… maybe so they can actually take it upon themselves, to go and do that”.(049, pregnant relapser)

The idea of a cost calculator, allowing women to see how much money they have saved by staying smoke-free and allowing women to input personalised saving goals tailored to their own wants/needs, was especially liked:
“What’s really helping me is that figure, that cost of what I’ve saved”.(046, pregnant ex-smoker)

Links to free distraction resources, such as games and quizzes, were also positively viewed. Women suggested needing more information specifically on postpartum weight management and mood (postpartum depression) in terms of supporting them to also remain smoke-free. Other suggestions were made that might improve intervention engagement, such as a logon to a website that recognised how many days women had been smoke-free and the ability to personalise the site by adding photos or linking to a text message support system.

#### 3.3.2. Digital/Text Message Support

The idea of receiving regular supportive text messages postpartum was generally liked, although not unanimously. To be most acceptable and helpful, women felt that text messages needed to start either during pregnancy, or as soon as the baby was born, and the messages needed be tailored, i.e., based on the age of the baby, and targeted towards supporting women to stay stopped from smoking, or encouraging new cessation attempts for those who may have relapsed:
“If you get a text message, everyone is straight on their phone. You know you hear an iPhone go off in like (place) town or something, or in a supermarket, and it’s all dinging and everyone’s like oh, is that my phone?! So as I said, really useful”.(044, pregnant ex-smoker)

Women and partners suggested a preference for texts with tailored content, e.g., using personal names and offering practical advice such as relapse strategies:
“Yeah I think using the person’s name is definitely—I mean there’s so much you know, we get so much kind of stuff marketed at us in this day and age, sort of digital marketing and that so I think yeah, I think definitely, you’d want probably a personal approach, you know. There’s no mistake, you know you’re getting it through text message onto your phone and it’s speaking directly to you so yeah, I think that probably is important to personalise the message”.(022, partner to pregnant relapser)

It was suggested that texts need to be very regular, perhaps daily to start with, but that decreasing frequency over time would be appropriate:
“I suppose once you’ve got into it a little bit and you feel like you don’t need as much help, then I suppose that could start to then tailor down. And that could tie in quite a bit with the relapsing part, so to speak, I suppose because if then you feel like you’re not doing well, you could then put on the like reply to the text message saying ‘oh I’ve had a relapse’ and then could then set you back to the three, four day thing but then if you’re doing well that might… maybe you could like do it every sort of five days to a week maybe”.(021, pregnant relapser)

It was important that women felt in control. They wanted the ability to stop text messages on demand. They also wanted to be able to increase or decrease the frequency of texts dependent on their individual needs. Brief texts were preferred, offering snippets of support or information. It was felt that texts may give links to further information, which could be followed up as and when women had the time.

### 3.4. ‘How Long’ Should Support Continue for?

It was thought to be important that the positive praise and tailored relapse prevention advice was followed up and reiterated. Having recognition of the achievement of staying smoke-free was important:
“The midwives have tick boxes, don’t they, they have to talk to you about and even if it was just a tick box every time you went, the midwife mentions it again—that would be helpful”.(050, postpartum ex-smoker)

Health visitors in the UK have a remit to support and care for women and families postpartum. They were felt to be well placed to discuss the benefits for babies and other family members of living in a smoke-free environment following pregnancy, and for offering support or referral to partners or other family members who were smokers, if that was applicable:
“when you see your health visitor, they support you with that as well because they give you information and support on like your breastfeeding and sleeping and other things to do with your child so they could like roll that into one and have the smoking support with that, then I think a lot of women might take to it”.(FG 053-054, pregnant ex-smoker)

### 3.5. Potential Intervention Components

#### 3.5.1. Relapse Prevention ‘Kit’ 

We proposed and asked for feedback on the idea that women might receive a ‘relapse prevention kit’. This would be followed up with tailored advice and support from a health visitor at the usual care postpartum at home visit (~10–14 days after the birth). Women generally liked the idea, although there was some concern about being reminded about smoking that might prompt cravings. Women particularly liked the prospect of being given ‘free gifts’ such as a voucher for a local coffee shop.

More information and advice specifically on the immediate post-birth period was thought to be helpful. This needed to be very positive and supportive, recognising the huge change in identity that women undergo as new mums:
“something that’s for them. You know, that’s what they need, it’s something for them and they know there’s help and they know that there’s support. So a little help pack. Mama’s Little Help Pack or Help Kit—that, I think that would go really well because it’s all personal”.(044, pregnant ex-smoker)

#### 3.5.2. Partner Involvement

Women thought it was important to involve partners, who were often perceived to be overlooked by health professionals during pregnancy and postpartum. Despite this, partner smoking status was recognised as being key to mothers’ ability to stay smoke-free after the birth:
“I think the best thing a partner can do is quit as well really because after I’d had my oldest child, he was still smoking and he was quite good at first and would say no I’m not, and I’d say oh just give me a cigarette and he’d say no, no, no I’m not but eventually I just pestered him when we were out one day and he gave in and that’s how I started again”.(FG 023–026, postpartum ex-smoker)

However, engagement with partners was raised as a difficulty. A tailored leaflet was felt to be potentially acceptable and helpful, as this would remove the need for women to have to try and address partner smoking herself, which could cause friction in the relationship. A leaflet for partners was considered important because partners were perceived as needing advice about how best to support the mum to stay smoke-free:
“he is very good at reeling facts off at you, but it’s the emotional support that you sometimes need. I knew, you know, people die of lung cancer. I knew that you can get heart failure, I knew of this, that and the other, but that emotional support is missing”.(034, postpartum ex-smoker)

Importantly, it was felt that the leaflet for partners could also promote partner smoking cessation and/or relapse prevention. Information for partners could be quite hard hitting, giving factual information about the harms of smoking around mums and babies:
“I think that’s quite a strong right approach actually. I mean they say when she was born you’ve got a new name ‘dad’, I mean that’s something important, isn’t it, it’s like a big thing. Isn’t it time you got rid of the other one? The dirty cigarettes and ashtray”.(052, partner)

Partners also supported this approach, and highlighted the need not just for smoking cessation support, but for information and support about how to support the mum to remain abstinent:
“even if it’s just to help them help the mum kind of quit, but it is just having that someone to talk to about how they do that, or how they help them quit or how they stand by them”.(051, partner)

#### 3.5.3. Social Support

Input from others—live chat, quotations or testimonials—were thought to be supportive and helpful components of a potential intervention:
“I like the Facebook chat thing, things like that, people talking about it on there would probably be more beneficial. People that are going through it”.(042, pregnant ex-smoker)

The idea of a self-help support group (‘parents forum’) was liked. Women appreciated being able to chat with others in the same situation. Although some women liked the idea of meeting up face-to-face, others mentioned how difficult this could be in the postpartum period, and also the attractiveness of online social support due to the relative anonymity:
“I think something maybe online more than in person. I think definitely, you definitely feel as a woman who’s gone back to smoking after having a baby, you do feel a lot of stigma, you do feel a lot of judgement towards yourself and I think maybe going somewhere where you’re in a group physically, you may feel more judged than say if it was online where it’s more anonymous”.(001, pregnant ex-smoker)

#### 3.5.4. E-Cigarettes

Content that gave advice and information on e-cigarettes was controversial. Many women stated that they knew e-cigarettes are likely to be much less harmful to health than continuing to smoke tobacco, yet also were very wary of using e-cigarettes. Concerns were raised about ‘continuing addiction’ to nicotine or a continued behavioural addiction:
“So personally, I don’t think promoting, because it’s still smoking, you’re not taking away that need for smoking, the habit of smoking or the nicotine dependency—you’re replacing it”.(034, postpartum ex-smoker)

Others were unsure about any possible long-term effects of use:
“I’m a bit concerned about the lack of long-term data we have on them about the levels of harm”.(FG 006–008, postpartum ex-smoker)

Some women in our sample had tried e cigarettes but had not got on with them. Others acknowledged that they might be useful for some people:
“for some that need it, yes, but i don’t like it. .... far safer?...........relatively harmless?. There’s still risks”.(Online anonymous feedback)

Most importantly, women, and also health professionals, wanted clear, precise information about e cigarettes, including advice on which ones to choose, which nicotine strength e liquid might be best and information about use while breastfeeding. Information on breastfeeding and e cigarette use was thought to be completely unavailable:
“Erm just knowing that the baby was being free from nicotine. Yeah I don’t know how much that is and how that could affect a baby. Erm, it’s kind of hard to say yeah, but it’s still not good that they’re getting nicotine, I mean even if you’re doing the e-cigarettes, I don’t know what the reduction in risks will be and what the risks are of nicotine through breastmilk for your baby are. So maybe information on that. Erm, I guess you just want to say still breastfeed even if you’re smoking, even if you’re vaping or if you’re not. You know it’s still the best way to feed your baby but erm, yeah I don’t know what the difference of nicotine exposure is through smoking cigarettes and vaping and you know what the effects on the baby are”.(055, postpartum ex-smoker)

#### 3.5.5. Incentives

Incentives for staying smoke-free were generally positively viewed by women:
“I think money would be an added motivation. Every little bit helps.”.(FG 006–008, postpartum ex-smoker)

Particularly so if the incentive was for the individual, rather than money or vouchers for the baby (although this was not unanimous):
“Yeah. Gift card. Why don’t you go and treat yourself because, yeah. I remember when after she was first born it was my birthday and [name]’s mum paid for me to get a massage and that was brilliant because I felt really properly spoilt, because you put so much of yourself into everybody else, you know, so yeah that would be a really good incentive I think. Then you can go and do your well-earned gift, or whatever it is, and feel good about yourself, which is what everybody needs, especially if they’re trying to stop smoking”.(004, postpartum ex-smoker)

However, incentives were less positively viewed by health care professionals, who expressed concern about how they might be funded and the longevity of the behaviour change that was incentivised:
“it doesn’t sit well with me that people are going to be… I think everybody knows the dangers, I think people do need to be reminded but I don’t think that giving rewards for that, like you say, is it just going to… ‘oh I’ll get my reward this time and then if I do it again, I can stop again and ooh do I get another reward like’”.(009–014 HCP FG)

The idea of self-incentives seemed to be more acceptable, and was supported and positively viewed by both health care professionals and women/partners:
“to put the money that you would have spent away and then look at how much you’ve got at the end of the month. That’s always a good one”.(050, postpartum ex-smoker)

### 3.6. Phase 3: Person-Based Intervention Development

The final phase of our study involved working closely with ten postpartum women, seven of whom had previously been involved in phase 2. We ran a prototype version of the intervention, including delivering the text message support system, and collected detailed qualitative feedback during four extended interviews with each woman, conducted over a three-month period. In this phase we iteratively developed the package of support and were reassured about its acceptability. The overall intervention was perceived positively, and all participants had extremely useful feedback and suggestions for improvements that will be incorporated into the final intervention. Overall, all participants felt that a smoking relapse prevention intervention was important, would be beneficial, and the proposed intervention pathway (Figure 1) was thought to be appropriate and missing no major elements.

Seeking feedback from women one to one enabled us to clearly understand the challenges of remaining smoke-free over time and as the postpartum period progressed. Regaining some normality to life following the disruption of childbirth, and encountering smoking triggers, was challenging:
“we’ve never smoked in our house, going for a dog walk is a chance for us to have a chat, catch up and have a couple of cigarettes on the way round. So actually, when I first gave up, going for a walk was one of my triggers for wanting to have a cigarette”.(058, postpartum ex-smoker)

Here, the participant emphasises the relational aspect of relapse triggers, since smoking with a partner whilst walking was a previously enjoyable habit. This demonstrates the enormity of the task of remaining smoke-free, in adjusting to a new ‘normal’ following childbirth, in coping with smoking triggers, in negotiating relationships where smoking had been an integral part, and in finding alternative ways to relax and cope with the stresses and strains of new motherhood.

Focusing the intervention not just on supporting the mother, but offering support for partners, both to quit smoking themselves, and advising on how to help women stay smoke-free, was seen as absolutely essential:
“It’s mainly just the thing about the partners that’s going to be the most helpful and I think for me would have been the most useful. Erm, just knowing that you know, because then I could have supported him, you know, so, and maybe it’s a platform that you can refer to as a couple and again, it starts that conversation and if it’s got facts and figures on it, I think it’s more likely to, it would have helped me to convince him”.(060, postpartum ex-smoker)

Having support that continued well into the postpartum period that was reiterated by health professionals, at every contact, was supported:
“And then it’s recapped during like your midwife appointments and things like that. I think that’s quite a good way to have it, and the website’s there continuously as well. And that you’ve been informed about it from the start, which is quite good because it gives you a chance to get used to using it and all the different sort of benefits and features of it. And then I think that the Fourth trimester pack (relapse prevention kit), sort of after your baby’s first born, it can be given to you, and then they can go over it with you when they visit and stuff, I think that’s quite good… I think the first year because then it’s all your pregnancy and then a few months afterwards. I think that’s sort of good to keep it going until you sort of have had your baby and then it still encourages you once you’ve had your baby to stay quit”.(063, postpartum ex-smoker)

Each intervention component was discussed in detail following a period of time in which participants had ‘used’ the intervention in real-time. Overall feedback was positive. Where negative comments were received, we used these to develop and refine the intervention.


**Component 1—Intervention introduction, leaflets and initial advice from a healthcare professional**


Women were positive about the intervention name—‘BabyBreathe^TM^’—having a positive supportive association. Supporting feedback from phase 2 that the intervention should be initiated during pregnancy, women were positive about our prototype leaflet introducing the intervention, but felt strongly that this needed to be part of or to initiate a conversation with a healthcare professional:
“I think as long as they’re giving the leaflet out and talking to people at the same time”.(062, postpartum ex-smoker)

The need for supportive, evidence-based information on the importance of remaining smoke-free was emphasised, but it was also strongly suggested that leaflets could be easily lost or overlooked, and that the power of having information reiterated by a ‘credible source’ was important to incorporate into the intervention.


**Component 2—Partner support**


The idea of a separate leaflet or information source for partners was very well supported. Women felt that it would be useful to have something specifically designed for partners, as it was perceived that partners were often excluded. Support for partners was also positively framed:
“you’re kind of already coming from a positive place, aren’t you? Saying thank so much for being that guy, thanks for being that person and you’re given the factual information about kind of why it’s great that you are this person and that you’re doing this and why it’s good for mum and baby. Erm, and you know you’re kind of giving enough information about the programme so you’re saying, we’re not asking you to do it alone with your partner, we’re going to help you do this and this is what we’re going to do to help you do this”.(059, postpartum ex-smoker)

The partner leaflet included cessation support for partners where this was appropriate, but also advice for partners on how to support their partner on staying smoke-free.


**Component 3—Website/app**


Women were given time to engage with the website in real-time prior to and during the interview. They were asked to consider the format of the information being on a website or available as an app. Overall, women reacted positively to the website content and format. Some women appreciated the factual information, stating that it was important that everything was up to date and ‘evidence-based’:
“I think for me, and again I think this might just be personal choice, it is the stats information, you know the kind of health, second-hand smoke, you know all those kind of stats, SIDS, increased risk of SIDS—all that kind of information for me, really stands out to me because I know that it’s backed up with research”.(059, postpartum ex-smoker)

Women especially appreciated being able to access support in their own time and being able to personalise/tailor the support. For example, we received positive feedback on a health timeline that incorporated women’s own dates, and on a savings calculator in which women could input their own data:
“it was personalising it to you and it’s something you can come back to and I think it’s really good so you’re aware of what your trigger points are and when you’re, you know your difficult times”.(060, postpartum ex-smoker)

It was important to have both a website and an app available. Some women preferred one format over the other. Predominantly, women accessed support on their phones:
“But probably more so on my phone than on an actual laptop. Especially if I was, and I had managed to breast feed, it’s probably something that I would have done whilst feeding”.(058, postpartum ex-smoker)


**Component 4—Text message support postpartum**


Tailored digital support via the text message support system received positive feedback, having been experienced in ‘real-time’. This involved sending 16 text messages over a period of four weeks, with reducing frequency. Tailored options included a quiz and questions that participants could respond to, such as texting ‘help’ if extra support was required. Women described the usefulness of continued positive support through hard times:
“They were really good, they were good, yeah. Yeah they did help you. I mean I think they would help especially for like if you’re having a bad day and you can just try, because everyone’s on their phones, aren’t they, so just try and distract yourself”.(064, postpartum ex-smoker)

For some, the idea of digital support also was useful in terms of helping them to feel less alone:
“It’s quite nice to hear my phone. I was like oh who’s messaged me! Oh yeah, look. And I suppose maybe it’s because I’m quite rural so therefore I am at risk of being really quite isolated. That oh yeah, there’s someone there”.(058, postpartum ex-smoker)


**Component 5—BabyBreathe^TM^ box**


The final intervention component was a prototype mock-up of a physical kit, a box that women might receive around the time of the birth of the baby. This would incorporate more advice and information, as well as some ‘free gifts’ such as vouchers for Mums to use as self-rewards/treats. We also discussed including a piece of NRT gum, and advice and support to use e cigarettes as a lapse prevention tool:
“I think it would be a good approach because it’s sort of then feels like you’re erm getting additional support and it sort of like gives you a chance to look through things, if you’re not going to use the website or the texting or things like that, it’s quite good to have a box full of the information. And different gifts and things that encourage you”.(062, postpartum ex-smoker)

Although advice on e-cigarettes remained controversial and was not for everyone, women appreciated knowing that they had alternatives they could turn to rather than returning to smoking:
“it’s always good to know what’s available because I mean like I was saying earlier, going back onto something like patches is so much better than going back onto cigarettes because then it’s only the nicotine you’re getting and it’s not anything else, is it? And I think just being aware of what things are really good. And obviously all the information that you guys have got, you always talk about e-cigs as well so I think it’s good then you’ve got both sides, haven’t you?”(060, postpartum ex-smoker)

It was emphasised that women needed very clear information, and that there was much confusion around the use of nicotine postpartum:
“Just explaining what… yeah. What are e-cigarettes, what are the different… the fact that you can have 0% strengths, you can have different variations of nicotine maybe. Yeah I think for a lot of people they might go back to cigarettes rather than that because that all just seems so confusing and complicated if you’ve never used any kind of nicotine replacement”.(066, postpartum ex-smoker)

Overall, the intervention pathway and combined package of support received positive feedback. It was important to have different elements that women could use in combination, or pick and choose from, tailoring the support required to suit their own needs:
“I think it would make you feel very supported. Yeah, definitely. A combination of everything, the website, the texts. Erm, the app”.(066, postpartum ex-smoker)

## 4. Discussion

Our three-phase mixed methods intervention development study demonstrates a process for utilising a theory-driven approach to define intervention components, that were then refined and initially ‘tested’ for real world use and acceptability over two qualitative phases of data collection and analysis. In our focus group and interview feedback phase, we gathered information on ‘who, what form, how long and which components’ might constitute an acceptable, helpful and feasible intervention. This enabled us to develop the intervention pathway and components, drawing on our findings to identify midwives or health visitors during pregnancy as being a ‘credible source’ to introduce a postpartum relapse prevention intervention. Women wanted evidence-based information that they could access towards the end of pregnancy, to prepare for the high risk (for smoking relapse) immediate postpartum period. It was also felt that the intervention could really ‘kick in’ at this time, with support offered face-to-face by a health visitor. There was a high level of acceptability for digital support. Women had positive feedback on a prototype website and were enthusiastic about using an app which they might access in their own time. Tailored elements were particularly appreciated, including a cost-saving calculator and a health calculator, so that women might be able to see, for them and for their baby, the benefits of staying smoke-free over time.

Women in our phase two data collection were unsure about the use of digital text message support (from birth), emphasising the importance of tailoring and being able to opt-out. In contrast, when we actually tested the text message support system in phase 3 in ‘real-time’, feedback was very positive. Particularly, text messages were thought to facilitate support to women at particularly risky times for smoking relapse, such as returning to work or social situations in which there was an association with smoking behaviour. The idea of a ‘relapse prevention kit’, in the form of our ‘BabyBreathe^TM^ box’ that gave more advice and free gifts at the end of pregnancy, also received positive feedback. This is an approach that has been successfully used for smoking cessation in the general population of smokers [22,24], with our findings also suggesting that this may be a promising approach for postpartum relapse prevention. Overall, the ability to tailor and personalise all parts of the intervention was crucial. It was thought important to provide evidence-based information on smoke-free motivators, including thinking of the baby, concerns of second- and third-hand smoke, information on health statistics and positively framed praise and support. Partner support was felt to be critical to the likely success of an intervention, and it was noted that the important role of the partner was often lacking acknowledgement in most areas of postpartum support.

Stress, guilt, perceptions of judgement, isolation and struggling with a non-smoking identity alongside adjusting to a new mothering identity were important factors that an intervention needed to address, as has been suggested in previous qualitative work [2,4]. We also found entrenched confusion around the evidence on e-cigarettes, and a real need to provide evidence-based advice on safety and harm reduction in the postpartum period. This might simply include wider dissemination of existing information resources that were not known about (e.g., resources produced by the smoking in pregnancy challenge group [25]).

The intervention we present here includes self-help, tailored options and pharmacotherapy for relapse prevention, which is the only potential intervention demonstrating some evidence of effectiveness in a recent Cochrane review (although studies were generally of low quality, the certainty of the evidence was moderate–low, and it was concluded that more research is needed). The BabyBreathe^TM^ intervention has been developed working with women, partners and health professionals. The person-based approach has been used in other settings in developing digital health interventions [14], and, we hypothesise, is likely to result in an intervention that ultimately is more acceptable, feasible to deliver and effective at supporting sustained smoking abstinence postpartum. Although effectiveness has yet to be demonstrated (and will form the next phase of our work), our intervention aligns with current NHS policy to address smoking and relapse prevention as part of the prevention agenda within the NHS long term plan [26], and the ‘making every contact count’ agenda [27]. Our BabyBreathe^TM^ intervention is in line with the NHS cancer strategy, which places a major emphasis on prevention [15], both for individuals and in terms of preventing generational transmission of unhealthy behaviours.

Drawing on qualitative feedback, our intervention includes varied and tailored self-help options delivered initially in pregnancy and into the postpartum period. Crucially our intervention includes support for partners, which has not typically, consistently or reliably been included in postpartum relapse prevention interventions, but is a critical influence [28]. The intervention proposes to test the prophylactic use of NRT for women to self-administer if they experience the return of nicotine cravings postpartum. This is a novel and so far untested approach to postpartum relapse prevention. Additionally, support and information to access and use e cigarettes to cope with urges to smoke will be an important aspect of the intervention. Although qualitative feedback on NRT and e cigarettes was mixed and there remain concerns and fears about the use of e cigarettes, we found, in line with recent evidence [25], a clear need for factual information on the safer use of nicotine tailored to the postpartum period.

A strength of our work is the incorporation of evidence from different sources, triangulating systematic review evidence with qualitative data gathered in phases two and three. We recommend this approach to intervention development, as the triangulation of data sources is essential to the development of a meaningful, acceptable and feasible to deliver intervention. Being theory-driven ensures that the intervention can be clearly described and evaluated in future testing. However, a possible limitation of our work is the primarily qualitative nature of the intervention development, with data drawn from a purposive sample. While this is a strength of the person-based approach to intervention development [14], it is also possible that we did not capture a broad enough range of views. We did not collect detailed demographic data from participants in order to maximise participation, as the focus of our work was on intervention development. This is a limitation however, and we are aware that our sample did not capture a broad range of views across women and partners of varying ethnicities. Seven of our phase 3 sample had also been involved earlier in phase 2. In hindsight, this was a limitation as women may have unduly positively evaluated aspects of the intervention that they themselves had suggested in phase 2. Clearly, further testing of the intervention is necessary to establish effectiveness and cost effectiveness, and to assess generalisability to a wider population of postpartum women. Our approach to analysis was rigorous and systematic. One researcher (TB) coded all qualitative data and a second researcher (CN) checked analysis for consistency and applicability. Further, we undertook a proportion of independent double coding, improving the reliability and trustworthiness of the coding. Although given increased time and resource we might ideally have double coded all qualitative data, our approach to team working and iterative discussion of analysis as it developed gives us reassurance that the coding is sufficiently valid and reliable for us to base our intervention development upon.

Practically, our proposed intervention has been developed working with the population that it ultimately targets, and also incorporating feedback from healthcare professionals involved in the care of women throughout pregnancy and postpartum. This, we hypothesise, maximises its likely future utility, as well as ensuring that it is clinically and contextually appropriate and can run alongside usual care processes at low cost to the health service. The intervention is scalable and potentially adaptable to other cultural contexts and healthcare systems worldwide. The intervention is theory-driven, being based on health behaviour change theory, and with each component clearly mapped to specific behaviour change techniques within the behaviour change technique taxonomy v1 [12,16]. This ensures a clear pathway to evaluation of both the overall intervention, and its constituent components. Following the process of theory-driven, mixed methods intervention development outlined in this paper, our next step will be to evaluate the intervention as part of a definitive randomised controlled trial.

## 5. Conclusions

Developing a relapse prevention intervention that may ultimately improve postpartum relapse rates was a complex process, involving three key phases. Following a phase of systematic reviewing of intervention behaviour change techniques [16], we gathered qualitative feedback from pregnant and postpartum relapsers and ex-smokers, partners and healthcare professionals to define who might deliver an intervention, in what form, for how long and with what potential components. Using a person-based approach, the final phase of our work tested a prototype intervention, enabling us to develop an approach suitable for testing in a definitive trial.

## Figures and Tables

**Figure 1 ijerph-16-01968-f001:**
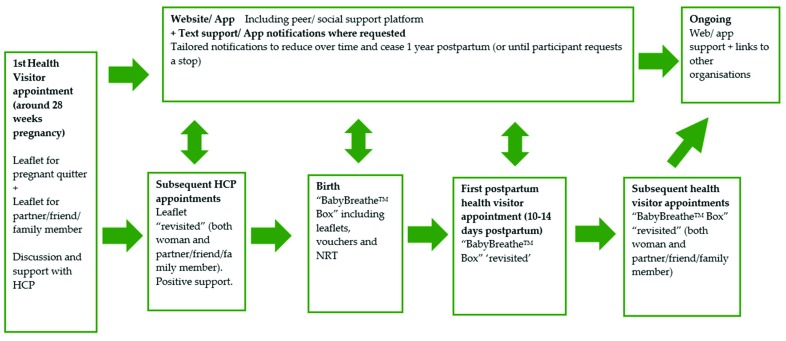
‘BabyBreathe’^TM^ intervention pathway.

**Table 1 ijerph-16-01968-t001:** Qualitative study participants.

Participants	Interviews Completed	Online/Email Feedback
Postpartum relapsers	7	2
Postpartum ex-smokers	16	6
Pregnant relapsers	5	0
Pregnant ex-smokers	9	4
Partners	7	2
Did not specify	0	4
Health professionals (Midwives, Health Visitors, Stop smoking advisors)	12	0
	**56**	**18**
Total		**74**

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
