# Peer review of "Development of a Complex Intervention for the Maintenance of Postpartum Smoking Abstinence: Process for Defining Evidence-Based Intervention"

_ijerph, 2019, doi:10.3390/ijerph16111968_

Round 1

Reviewer 1 Report

The paper is very well written and includes all major components of postpartum relapse that need to be addressed by a prevention intervention.

The Baby Breathe Box is a good name for an incentive, it is not clear if this box is always to be individualized or will contain standard components.

In our focus group research with both patients and care providers we had similar findings: finding new coping methods, message needs to be reiterated in pp period, message given at every contact, suggestions for alternative behaviors for relapse triggers.

It may be important to place more significance on second and third hand smoke exposure to newborns occurring with postpartum relapse.

My only suggestion is to include the complete names spelled out for the Com B framework in line 72 and Tidier framework in line 105.

Author Response

Thank you for the positive feedback. We are heartened to hear that the reviewer found similar results in their own work. We note the suggestion to include more information on second and third hand smoke exposure in the eventual intervention. However we do not note any suggested change to the submitted manuscript in the light of this suggestion.

My only suggestion is to include the complete names spelled out for the Com B framework in line 72 and Tidier framework in line 105.

These changes have now been made on the revised manuscript.

Reviewer 2 Report

Thank you for the opportunity to review the manuscript titled “Development of a complex intervneion for the maintenance of postpartum smoking abstinence: Process for defining evidence-based intervention”. This manuscript examines the postpartum smoking relapse intervention program development process, including needs assessment and the initial formative evaluation.  Overall, this manuscript is very well written with clear study objective, significance and qualitative study results demonstration. However, some minor issues should be addressed to improve the quality of the manuscript. My review of the manuscript and recommendations to the authors are enclosed. 

1.       Page 2, line 72, what’s the full spell of the COM-B framework? Line 86, what does the COREQ mean? Same on page 3 line 105, what’s TIDieR framework? Please use full name at the first time they were mentioned. 

2.       Although the authors stated they recruited 64 women in order to seek max variations, this statement lacks of evidence, and the readers may be still confused about whether the sample were representative, e.g., where and how these participants were recruited. Also, were participants all from a nearby neighborhood, or same city in UK or even the same hospital or clinic? Were participants from the same clinic as the health professionals?

3.       The demographic characteristics for all 74 participants should be reported in Table 1. In addition, the basic smoking behavior information should also provided for those current or ex-smokers.

Because the different postpartum stage has different relapse rate. The women in their 6 or 12 month postpartum viewed the relapse (including their craving status) different from the women in their first postpartum month. Also, the women in low SES have been reported that they had different views on the triggers of postpartum relapse and specifically, they had different barriers or acceptability for the use of website/internet/app or text based intervention, as the cost of wifi and text fee may not applied.   

4.       Figure 1 indicated that the “Baby Breathe” targeted on all three prenatal, birth and postpartum stages. Also, in the discussion section (line 589-590), the authors also stated the present study included varied and tailored self-help option delivered in pregnant and into the postpartum period. However, I thought the focus of this study was to develop a POSTPARTUM smoking relapse intervention, and all participants gave the feedback to define the intervention components postpartum. If the intervention did include all three stages and this study only focused on part of it, e.g., the postpartum stage, the authors should clarify.

5.       The authors addressed feedback from participants about their attitudes of integrating the app or tailored text based message into the intervention and their “real time” intervention experience. It will be better if the authors summarized clearly the intervention components first based on the phase 2 when describing the phase 3 on page 10. In addition, they may provide description for each component in some details. E.g., how many text messages were sent for each participant averagely? When these messages were sent? How the text messages were tailored or customized in the intervention? What did the evidence-based information the leaflet contain to improve the partner or social support?

6.       It seems all participants provided positive feedback for each intervention component. This result may be because seven of ten postpartum participants were from the phrase 2, which is not an overlook bias. Also, did the participants have any negative feedback or concern about any component of the program?

7.       Line 575 states the intervention includes self-help, tailored options text and pharmacotherapy for relapse prevention. What the pharmacotherapy this intervention provided for participants? Where was NRT mentioned in the text? The authors indicated that there was a discussion about NRT gum with participants.

Author Response

We thank the reviewers for their kind and supportive comments.  We have responded in full to all of the suggestions as outlined below:

1.       Page 2, line 72, what’s the full spell of the COM-B framework? Line 86, what does the COREQ mean? Same on page 3 line 105, what’s TIDieR framework? Please use full name at the first time they were mentioned. 

These changes have been made.

2.       Although the authors stated they recruited 64 women in order to seek max variations, this statement lacks of evidence, and the readers may be still confused about whether the sample were representative, e.g., where and how these participants were recruited. Also, were participants all from a nearby neighborhood, or same city in UK or even the same hospital or clinic? Were participants from the same clinic as the health professionals?

We have clarified the sampling approach, parameters and limitations in the manuscript. This was a purposively selected qualitative sample. Our aim was to maximise the likely variation in views by sampling across a broad range of characteristics (e.g. ex-smoker, relapsed to smoking, pregnant and postpartum women). We would not claim this sample to be representative in a statistical sense, but are certain that the sample incorporated a wide range of perspectives, and therefore the results have transferability to similar populations. Participants were sampled from one county of the UK. This is a limitation that we have included in the revised manuscript. The county has a broad spread of population characteristics, and pockets of deprivation. The pragmatic limitations of the research prevented sampling from other geographical areas. This is a limitation of our sample that does not capture ethnic variation, as we report in the limitations section of the paper. As a research team we did not work as health professionals within clinics used for recruitment purposes. Clinical support for recruitment was via the Clinical research networks. However the Health Professionals included in Focus Groups came from the same health services as the patients, thus they were able to discuss implementation of the interventions within relevant service settings.

3.       The demographic characteristics for all 74 participants should be reported in Table 1. In addition, the basic smoking behavior information should also provided for those current or ex-smokers.

Because the different postpartum stage has different relapse rate. The women in their 6 or 12 month postpartum viewed the relapse (including their craving status) different from the women in their first postpartum month. Also, the women in low SES have been reported that they had different views on the triggers of postpartum relapse and specifically, they had different barriers or acceptability for the use of website/internet/app or text based intervention, as the cost of wifi and text fee may not applied.   

We thank the reviewer for these points. We agree these are important considerations for us to capture in testing the effectiveness of the intervention in the next phase of our programme of work. In the development study reported in this manuscript we did not collect this detailed demographic and smoking data. This was a concious decision in order to maximise recruitment to the qualitative intervention development study. Relapse to smoking following childbirth is a sensitive and potentially stigmatised topic. We were keen to maximise recruitment, and therefore careful to point out to participants in all our recruitment materials that we were not interested in collecting data on their personal experiences and patterns of smoking, but wanted to gather data specifically on their views on ideas for an intervention. Although the lack of demographic data is a limitation of our study, we feel that this approach enhanced our recruitment and ensured that we had wide participation, especially including relapsed smokers and partners of women who are often ‘hard to reach’ and therefore not often included in research. In our next study we fully intend to capture a range of detailed demographic and smoking related behavioural outcomes at both baseline and follow up, to give us the opportunity to explore intervention engagement and effectiveness according to individual characteristics.

4.       Figure 1 indicated that the “Baby Breathe” targeted on all three prenatal, birth and postpartum stages. Also, in the discussion section (line 589-590), the authors also stated the present study included varied and tailored self-help option delivered in pregnant and into the postpartum period. However, I thought the focus of this study was to develop a POSTPARTUM smoking relapse intervention, and all participants gave the feedback to define the intervention components postpartum. If the intervention did include all three stages and this study only focused on part of it, e.g., the postpartum stage, the authors should clarify.

This is correct that the intervention is focused on postpartum relapse prevention. However it is also correct that the intervention begins during pregnancy and is delivered around the time of birth as well as in to the postpartum period. This is because our developmental work demonstrated that it is essential that the intervention begins during pregnancy in order to prepare women for the immediate postpartum period. We apologise if this was not clear in the manuscript and have carefully checked the wording to clarify this point.

5.       The authors addressed feedback from participants about their attitudes of integrating the app or tailored text based message into the intervention and their “real time” intervention experience. It will be better if the authors summarized clearly the intervention components first based on the phase 2 when describing the phase 3 on page 10. In addition, they may provide description for each component in some details. E.g., how many text messages were sent for each participant averagely? When these messages were sent? How the text messages were tailored or customized in the intervention? What did the evidence-based information the leaflet contain to improve the partner or social support?

These details have been added within each section for the phase 3 feedback under each of the intervention components.

6.       It seems all participants provided positive feedback for each intervention component. This result may be because seven of ten postpartum participants were from the phrase 2, which is not an overlook bias. Also, did the participants have any negative feedback or concern about any component of the program?

Overall feedback was positive, but of course there was some negative feedback. Particularly following phase 2, we used this feedback to make amendments to intervention components. This has been clarified.

7.       Line 575 states the intervention includes self-help, tailored options text and pharmacotherapy for relapse prevention. What the pharmacotherapy this intervention provided for participants? Where was NRT mentioned in the text? The authors indicated that there was a discussion about NRT gum with participants.

This point is addressed in the discussion of feedback in phase 3, for component 5, the BabyBreathe box. In this box we proposed including a piece of NRT gum.